# Extracellular Vesicles and MicroRNA in Myelodysplastic Syndromes

**DOI:** 10.3390/cells12040658

**Published:** 2023-02-19

**Authors:** Mathieu Meunier, David Laurin, Sophie Park

**Affiliations:** 1Department of Hematology, University Grenoble Alpes, CHU Grenoble Alpes Hospital, CEDEX 09, 38043 Grenoble, France; 2University Grenoble Alpes, Inserm U 1209, CNRS UMR 5309, Équipe Epigenetics Regulations, Institute for Advanced Biosciences, 38000 Grenoble, France; 3Etablissement Français du Sang, 38700 La Tronche, France

**Keywords:** extracellular vesicles, miRNA, non-coding RNA, myelodysplastic syndromes, exosomes, microvesicles, bone marrow niche

## Abstract

The bone marrow niche plays an increasing role in the pathophysiogenesis of myelodysplastic syndromes. More specifically, mesenchymal stromal cells, which can secrete extracellular vesicles and their miRNA contents, modulate the fate of hematopoietic stem cells leading to leukemogenesis. Extracellular vesicles can mediate their miRNA and protein contents between nearby cells but also in the plasma of the patients, being potent tools for diagnosis and prognostic markers in MDS. They can be targeted by antisense miRNA or by modulators of the secretion of extracellular vesicles and could lead to future therapeutic directions in MDS.

## 1. Introduction

Myelodysplastic syndromes (MDS) are a heterogeneous group of clonal diseases characterized by ineffective hematopoiesis due to dysmyelopoiesis, progressive cytopenias, and increased risk of developing acute myeloid leukemia (AML) in elderly patients. Disease evolution leads to the emergence of mutant genetically instable clones and transformation to AML in approximately 30% of the cases [1]. Prognosis is determined by the international prognostic scoring system (IPSS), recently refined as IPSS-R, allowing classification in two main stages: low- and high-risk MDS [2,3], according to the risk of progression to AML.

After diagnosis and risk status classification, additional factors determine the treatment the patient will take. For low-risk (LR) MDS, if asymptomatic, the patient will likely undergo a period of watchful waiting. However, patients with LR MDS and symptomatic cytopenias (neutropenia, anemia, and thrombocytopenia) will typically receive supportive care, namely, erythropoiesis-stimulating agents such as erythropoietin (EPO) and/or red blood cells/platelet transfusions for supportive care or recombinant granulocyte colony stimulating factor (G-CSF) therapy for patients who are neutropenic. Patients who have an excess of ring sideroblasts (RS) and low to intermediate risk who receive red blood cell (RBC) transfusions are also able to receive luspatercept (approved by the Food and Drug Administration [FDA] and the European Medicines Agency [EMA] in April 2020 for MDS) if they had previously failed, were intolerant to, or ineligible for EPO. Patients who receive transfusions are also likely to receive iron chelators and anti-fibrinolytic drugs (to help with blood clotting).

Most patients with higher-risk disease receive active therapy, with hypomethylating agents being the most commonly used modality, but allogeneic hematopoietic stem cell transplant may also be used for fit patients. Most patients with MDS, regardless of risk status, will require RBC transfusions at some time during their illness, with some patients becoming transfusion-dependent (TD). The transfusion burden is associated with the prognosis in patients with MDS. Those patients are susceptible to transfusion iron overload, which can adversely affect hepatic, cardiac, and endocrine functions. LR MDS patients will eventually progress to either higher-risk (HR) MDS or AML.

Several studies demonstrated that MDS were triggered by clonal abnormalities arising in the hematopoietic stem cells. Karyotype abnormalities are present in around 50–60% of de novo MDS and in 80 to 90% of secondary MDS [4,5]. Gene mutations are present in more than 90% of patients. The most frequent mutations identified in MDS occur in genes encoding components of the spliceosome or in epigenetic regulators [6,7]. Myelodysplastic syndromes result from the sequential acquisition of mutations in hematopoietic stem cells that will be responsible for clonal expansion and damage to myeloid differentiation.

In the last few years, the importance of the bone marrow (BM) microenvironment in the physiopathology of MDS has been highlighted. Normal and pathological hematopoiesis takes place in a specialized niche within the bone marrow microenvironment. A majority of the cells constituting this niche are derived from the bone marrow mesenchymal stromal cell (MSC), but there are also numerous other cells different from the hematopoietic cells defining the bone marrow niche. The recent studies in murine models show that abnormalities of MSC contribute to the physiopathology of MDS.

In this review, we will summarize the studies establishing a causal relationship between deregulation of the hematopoietic niche and MDS pathogenesis, stressing on the central role of mesenchymal stromal cells, their crosstalk between MSC and hematopoietic stem cells (HSCs), and especially, on the extracellular vesicles (EVs) and their contents, the microRNA (miRNA). In a second part, we will stress the importance of EVs and miRNA as diagnostic and prognostic tools. In the last part, we will finish on the therapeutic avenues using these innovative concepts of EVs and miRNAs.

## 2. Crosstalk between the Bone Marrow Niche and Hematopoietic Stem Cells: Role of Extracellular Vesicles

### 2.1. The Bone Marrow Niche and the Mesenchymal Stromal Cells

The two main architectural scaffolds of the bone marrow are the bone and the vessels. Classically, an endosteal niche in contact with bone tissue and a vascular niche in contact with endothelial cells are described. Lord et al. [8] observed that the hematopoietic stem cells were localized in the sub-endosteal sector in contact with the bone, suggesting that the osteoblast cells could regulate hematopoiesis. Indeed, osteoblasts are able to produce granulocyte colony-stimulating factor (G-CSF) [9] and interleukin 6, promoting the preservation of the immature character of hematopoietic stem cells [10]. In fact, the osteoblast was the first cell in the hematopoietic niche described to be able to influence HSCs. The osteoblasts that mainly make up the endosteal niche are derived from the same cell, the mesenchymal stromal cell.

Considering the localization of HSCs near blood vessels, attention has also focused on the mesenchymal stromal cells that surround the blood vessels at the level of the vascular niche. These perivascular mesenchymal cells express CD146 [11], CXCL12 [12], and Nestin [13]. CXCL12 positive reticular cells (CAR cells) adjacent to sinusoids are shown to co-localize with HSCs [12]. Subsequently, bone marrow MSCs that express Nestin are shown to localize around blood vessels in contact with HSCs and, furthermore, express high levels of stem cell factor (SCF) and CXCL12 [13]. These studies demonstrate that MSCs are one of the components of the perivascular niche for HSCs and that MSCs are ubiquitous within the hematopoietic niche and have a major role in the regulation of hematopoiesis. The perivascular stromal cells and endothelial cells that synthesize CXCL12 are the same cells and represent the primary source of SCF within the bone marrow [14]. Those cells produce multiple factors and are crucial components of the hematopoietic niche.

Mesenchymal stromal cells are a non-hematopoietic bone marrow cell population considered a key component of the hematopoietic microenvironment.

According to the international society for cell therapy [15], mesenchymal stromal cells are defined by three criteria:The ability to adhere to plastic,Expression of CD73, CD90, and CD105 with concomitant lack of expression of CD45, CD34, CD14 or CD11b, CD79α or CD19, and HLA-DR,The ability to differentiate into osteoblasts, chondrocytes and adipocytes in vitro. This definition was recently updated by the ISCT MSC committee, but it is recommended to mention the tissue-source origin of the cells to confirm evidence for stemness associated with a robust matrix of functional assays to demonstrate MSC properties [16].

Bone marrow is also constituted of other non-hematopoietic cell types such as osteoblasts, endothelial cells, pericytes, adipocytes, Schwann cells, and nerves and hematopoietic cell types such as macrophages, osteoclasts, and lymphocytes [17].

### 2.2. Exosomes and Microvesicles: Different Kinds of Extracellular Vesicles (EVs)

Microparticles and exosomes are classically separated by size criteria but there is actually a significant size overlap, and evidence of intra-cellular origin should be provided in order to stringently define exosomes [18] (Figure 1).

Large EVs or microvesicles are released during cell-surface budding, and their sizes range from 100 to 1000 nm in diameter. They are composed of an outer lipid bilayer that surrounds the inner content, composed of mRNA, miRNA, non-coding RNAs, proteins, and lipids. By contrast, smaller EVs or exosomes are derived from the endosomal membrane compartment. This leads to the formation of endosomes that contain intraluminal vesicles, named endosomal multivesicular bodies, which can release their contents in small exosomes after fusion with the plasma membrane into the extracellular space. Exosomes are defined by homogeneous membranous vesicles lined by a lipid bilayer and produced by the inner budding of the endosomal membrane during maturation of the multivesicular body (MVB). They are secreted by fusion of MVB with the cell surface. Their sizes vary from 50 to 150 nm in diameter. They contain a set of proteins, lipids, and nucleic acids (including miRNAs) that allow them to operate as cargo and signaling platforms for short-range or long-range delivery of information to other cells [19]. In cancer, they can modulate stromal, endothelial, inflammatory, and immune responses, thus, reprogramming the microenvironment that favors tumor progression and metastatic dissemination [20]. Exosomes, and more generally, small EVs (sEVs), are potent platforms for signaling and exchange of materials between MSCs and HSCs in the extremely promiscuous network of interactions of the medullary microenvironment (Figure 1) [21].

However, there is currently no consensus for the nomenclature of extracellular vesicles. Differential ultracentrifugation is the most widely used technique to separate and collect secreted vesicles. The different vesicles can be defined as a result of the centrifugation speed used: the microvesicles correspond to the pellet obtained at 10,000× *g* and the exosomes obtained at 100,000× *g* [22]. In addition, exosomes were conventionally described as having a cup-shaped morphology and a size of 50 to 100 nm under electron microscopy. However, this characteristic appearance was shown to be due to an experimental artefact, probably due to the drying process. Initially, a proteomic analysis of the different fractions obtained during the differential centrifugation steps was carried out. A categorization of extracellular vesicles was proposed, which could be applied to all sources of extracellular vesicles (cultured cells or biological fluids) [23]. Nevertheless, the origin and the production route are extremely difficult to determine on the vesicles themselves. The tetraspanin markers and others, for example, which were considered specific to exosomes, are, in fact, also present on microvesicles [24].

Moreover, it is difficult to separate exosomes from microvesicles technically. However, when studying the biological effects of the cell secretome in order to mimic the in vivo situation, we should evaluate the biological impacts of these different vesicles together.

**Figure 1 cells-12-00658-f001:**
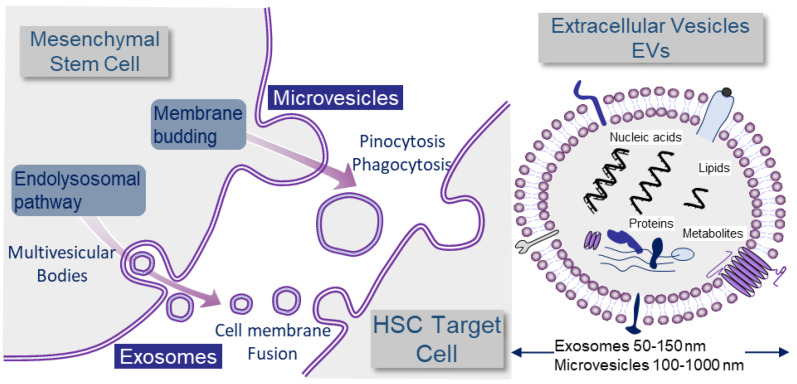
Definition of exosomes and microvesicles in the context of MSC to HSC communication. Large EVs or microvesicles are released during cell-surface budding, while smaller EVs or exosomes are derived from the endosomal pathway. All cells produce EVs. Here, we show the example of EV production by mesenchymal cells because they play an important role in hematopoiesis and are in close interaction with hematopoietic stem cells (HSC). EVs are an important means of communication between cells and deliver signals in the form of nucleic acids, including miRNAs, lipids, proteins, or metabolites. The mechanisms of non-viral EV transfer are not yet fully elucidated. EVs may have ligands capable of binding to corresponding receptors on target cells. These trigger signaling cascades and can induce receptor-mediated endocytosis. Despite the fact that the mechanisms by which receptor cells take up exosomes have not been fully elucidated, there are some studies demonstrating that integrins, lectins/proteoglycans, T cell immunoglobulin, and mucin structural domain protein 4 (Tim4) may contribute to cellular targeting specificity [25,26]. EV entry may also involve endocytosis mediated by clathrin-coated pits, lipid rafts, phagocytosis, caveolae, and micropinocytosis [27]. Finally, the entry of EV contents may also involve membrane fusion or macropinocytosis to spill their contents into the cytoplasm. Thus, bioactive molecules contained in EVs can also be transferred into the cytoplasm of the target cell and exert activity there. EVs can also simply be degraded, thus, becoming a source of nutrients for the recipient cells [28]. Moreover, EVs can mediate the interaction of secretory cells with the surrounding extracellular matrix (ECM). In mainly tumor contexts, EVs were shown to function also in long-distance communication.

### 2.3. MicroRNA

MicroRNAs (miRNAs) are short non-coding RNAs that regulate gene expression post-transcriptionally [29]. To date, more than 2600 human mature miRNAs have been identified, each with the potential to regulate hundreds of target genes [30]. MicroRNAs play key regulatory roles in all biological processes, including cell proliferation, cellular differentiation, cell cycle control, apoptosis, and angiogenesis. MicroRNAs can act as either oncogenes or as antioncogenes. MicroRNAs are important regulators of the differentiation and maintenance of hematopoietic stem cells, and changes in their expression levels can be correlated to the development of myeloid and lymphoid neoplasms.

Mature miRNA is 19–22 nucleotides in length, and its formation is a multistep process involving a large number of enzymes. The miRNA genes are first expressed as a long precursor by a polymerase II or III. This precursor, called pri-miRNA, has a polyA tail contributing to its stability. Then, this pri-miRNA will be processed by a protein complex composed of Drosha (an endonuclease) and its cofactor DiGeorge syndrome critical region 8 (DGCR8). This protein complex will cleave the pri-miRNA and generate an RNA of 60 to 100 nucleotides with a hairpin structure: pre-miRNA. The latter will be exported out of the nucleus to the cytoplasm by exportin 5. In the cytoplasm, the pre-miRNA will be processed by RNA III endonuclease (DICER1), which will cleave the hairpin structure and generate an RNA double strand of 21 to 23 nucleotides [31]. Subsequently, DICER1 will associate with other proteins, in particular with the Argonauts family (Ago1 or Ago2) and trans-activation response RNA binding protein (TRBP) to form the complex RNA-induced silencing complex (RISC), which will support double-stranded RNA. A helicase will separate the two strands of RNA and only keep one of the two, which will be called the mature miRNA [32].

Mature miRNAs are often suffixed “5p” or “3p” to denote the functional miRNA strand.

The strand with the 5′ uracil is called “the guide strand” and can be incorporated into the RNA-induced silencing complex (RISC), leading to the post-transcriptional regulation of target genes, while the other strand, called “the 3′ passenger strand”, undergoes rapid degradation. The target gene is silenced by mRNA cleavage at 10–11 nt upstream of the 5′-end of the guide strand. This cleavage is mediated by the activity of the Ago2 protein, one of the main components of the RISC complex. In most cases, miRNAs interact with the 3′-UTR of target mRNAs. The degree of complementarity determines what will happen: Ago2-dependent cleavage of the target mRNA or translational suppression [33]. Perfect complementarity allows Ago2-catalyzed cleavage of the mRNA strand. However, mismatches exclude cleavage and promote repression of mRNA translation. RISC can inhibit mRNA translation by interfering with the eiF4F complex, then the polyadenylase PAN2/3 and CCR4-NOT complex deanylate the mRNA, removing the m7G cap of the target mRNA. The decapped mRNA then undergo 5′-3′ degradation via the exoribonuclease XRN1 [34]. The miRNA incorporated into RISC complex (miRISC) can also repress a post-initiation stage of translation by inducing ribosomes to drop off prematurely [35]. In most cases, miRNAs interact with the 3′-UTR of target mRNAs; however, interactions of miRNAs with other regions, including the 5′-UTR, the coding sequence, and gene promoters, were also reported [36]. Currently, there are information resources (miRTarBase, TargetScan, mirDB, miRWalk, miRanda) that can predict in silico the genes targeted by miRNAs [37].

Here in this review, we will develop the role of extracellular vesicles secreted by the mesenchymal stromal cells and the role of miRNA cargoed by EVs in the physiopathology, diagnosis, prognosis and their possibility of modulation for therapeutic purposes, in myelodysplastic syndromes.

### 2.4. Medullar Microenvironment Contribution to MDS Physiopathology (Figure 2)

The physiopathology of MDS is complex. Clonal hematopoiesis and the microenvironment could play complementary roles. Raaijmakers et al. [38] have demonstrated in a murine model that the deletion of DICER1 in MSC-derived osteoprogenitors not only affected their differentiation but also resulted in the generation of myelodysplasia and secondary leukemia. These results demonstrate that specific molecular alterations in the bone marrow microenvironment could result in clonally impaired hematopoiesis. We have also demonstrated, as have other teams, that MSCs from MDS patients, compared with healthy subjects, have a lower level of expression of DICER1 and DROSHA [39,40,41]. Furthermore, Moiseev et al., using DNA sequencing on bone marrow from high-risk MDS patients, demonstrated that mutations in the miRNA processing genes were present in approximately 50% of the patients [42]. Moreover, overexpression of beta-catenin in murine osteoblasts was shown to cause aberrant activation of Notch and FOXO-1 signaling in HSC, resulting in AML transition [43,44]. These observations establish a causal relationship between deregulation in the bone marrow microenvironment and MDS pathogenesis.

The bone marrow niche also participates in the pathogenesis of MDS via a pro-inflammatory climate. The TLR4/S100A8/S100A9 axis, leading to the activation of the inflammasome, is triggered in HSCs by hematopoietic inhibitory myeloid-derived suppressor cells (MDSCs) [45,46]. The alarmin S100A9 triggers pyroptosis through the generation of reactive oxygen species, leading to assembly and activation of the redox-sensitive NLRP3 inflammasome and beta-catenin, assuring propagation of the MDS clone by DNA damage and lesions in the HSCs. Zambetti et al. identified a mesenchymal-niche-induced inflammatory signaling axis that results in genotoxic stress in HSCs in MDS [47]. Innate immunity and miRNA play also an important role in a subtype of MDS, 5q deletion syndrome. MDS with chromosome 5q deletion represents a distinct hematologic and pathologic subgroup defined by the 2016 WHO classification that is characterized by an isolated 5q deletion or del(5q) and specific features as the association of anemia with leukopenia and thrombocytosis. This syndrome is characterized cytogenetically by the loss of part of the long arm (q arm, band 5q33.1) of human chromosome 5 in myeloid progenitors [48]. The 5q deletion induces the loss of two miRNAs that are normally overexpressed in hematopoietic stem/progenitor cells (HSPCs), miR-145 and miR-146a. These miRNAs respectively target Toll-interleukin-1 receptor domain-containing adaptor protein (TIRAP) and tumor necrosis factor receptor-associated factor-6 (TRAF6). Knockdown of that miRNA or overexpression of TRAF6 in mouse HSPCs resulted in thrombocytosis, mild neutropenia, and megakaryocytic dysplasia, which are several characteristic features of 5q deletion syndrome [49].

The MSCs in MDS are “pathologic” [50]. They harbor clonal chromosomal alterations but are distinct from those commonly reported in HSCs [51,52]. In Blau et al.’s study [51], the most frequently found abnormalities in MDS-derived MSCs are 7q deletion, a deletion of chromosome 4 or Y, and the addition of chromosome 5. In this study, which also included MSCs from patients with AML, interestingly, patients with cytogenetic abnormalities within their MSC exhibited a poorer overall survival than patients with MSC free from cytogenetic abnormalities. In addition, certain cytogenetic abnormalities present in MDS-derived MSCs are said to be specific for a subtype of myelodysplastic syndrome [53].

MDS-derived MSCs show impaired osteogenic differentiation capacity, earlier senescence, altered clonogenic capacity of MSCs, altered methylation pattern, Osteopontin, jagged1 increased, kit ligand, and angiopoetin decreased [50]. Comparative gene expression profiling of MDS-derived MSCs versus healthy donor MSCs showed reduced expression of DICER1, Drosha, SBDS, and various miRNA, including miR-155, miR-181a, and miR-222 in MDS-derived MSC [41].

A permanent crosstalk takes place within the hematopoietic tumor niche between the hematopoietic tumor cells and their microenvironment, especially mesenchymal stromal cells [40,54]. These interactions participate in oncogenesis [38,55]. The intercellular crosstalk involves direct intercellular contact, the release of cytokines, growth factors, as well as extracellular vesicles, which contain different components, such as miRNA, mRNA, proteins, and lipids, which can regulate important biological processes in the targeted cells [56].

### 2.5. EVs Mediate miRNA to HSCs in MDS, Crosstalk between MSCs and HSCs into Bone Marrow Niche (Figure 2)

Vesicles are a means of intercellular communication between cells and, especially, between mesenchymal stomal cells and hematopoietic stem cells. Phenotypically, EVs by themselves can influence the fate of HSCs. Meunier et al. showed that small EVs from MDS-derived MSCs induced HSC apoptosis and can also promote an oxidative environment with increased ROS level and induction of DNA damage. Whole genome sequencing of healthy donor CD34^+^ cells incubated with sEVs from MDS-derived MSCs showed that sEVs from MDS-derived MSCs are able to induce a mutational signature found in cancers listed in the COSMIC cancer database (https://cancer.sanger.ac.uk/cosmic, accessed on 25 December 2022). The majority of mutations concerned the intergenic and intronic regions, and very few gene mutations were observed in the coding regions. Genetic mutations are not the sole explanations for the complete development of MDS, and those epigenetic modifications might be the soil for further addictive gene mutations [39]. It was suggested that MVs derived from human bone marrow MSC may act as mediators of cell-to-cell communication through miRNA delivery [57]. These transferred micro-RNAs could play an important role in the hematopoietic system.

Muntion et al. isolated microvesicles by two methods: ultracentrifugation and the Exoquick system, with the globally same characterization of EVs by electronic microscopy and CD63 and CD81 markers. They have shown that sEVs from MDS-derived MSCs could induce better clonogenic capacity of CD34^+^ cells. They clearly demonstrated by confocal microscopy that sEVs from MSCs are incorporated into CD34^+^ cells. MiR-10a and miR-15a are two of the most overexpressed miRNAs in the MSC-derived EVs from MDS patients. Those miRNA target cell cycle, TP53, and PI3K/AKT signaling. They suggested that the increased erythroid progenitor apoptosis seen in MDS could be mediated by MVs from the microenvironment carrying miRNAs acting on the TP53 pathway [40].

Our team confirmed that MDS-derived MSCs support an impaired hematopoiesis compared to healthy donor stroma. We confirmed the underexpression of DICER1 in low-risk human MDS MSCs via a total marrow flow cytometry technique [39]. This underexpression was responsible for an abnormal profile of miRNAs expressed by MDS MSC with 16 overexpressed miRs and 7 underexpressed between MDS MSCs and from healthy subjects. We have demonstrated a miRNA of interest that is overexpressed in MDS MSC, miR-486-5p. This miR is implicated in the acutization of MDS in acute leukemia in patients with trisomy 21 [58] and in the blast transformation of chronic myeloid leukemia [59]. We confirmed that this miR could be secreted into small vesicles from MDS MSCs and incorporated into HSCs. Using RNA sequencing of CD34^+^ cells from a healthy donor overexpressing miR-486-5p, we found that the upregulation of this miRNA led to the activation of TNFα, innate immune, and inflammatory pathways [39].

Moreover, another study from Saitoh et al. [60] demonstrated also the intercellular communication into bone marrow niche via EVs. They observed that the EV content of miR-101 was lower in EVs from bone marrow MSCs from high-risk MDS patients than bone marrow MSCs from healthy donors and low-risk MDS patients. Interestingly, they demonstrated that the level of miR-101was, on the contrary, higher in bone marrow MSCs from high-risk MDS patients compared to bone marrow MSCs from healthy donors and low-risk MDS patients. They suggested that some specific miRs from bone marrow MSCs may be actively selected into EVs (intra-EV accumulation), and this phenomenon could be linked with the increased blast number observed in patients with high-risk MDS.

**Figure 2 cells-12-00658-f002:**
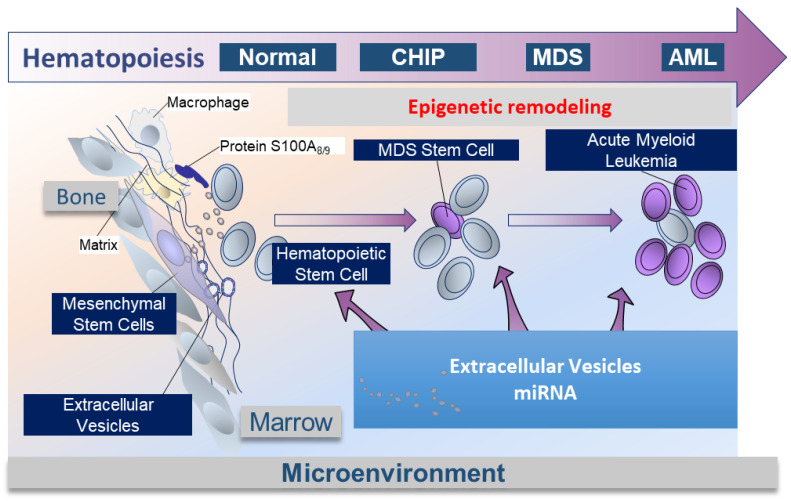
Schema of the role of EVs and miRNA in the physiopathology of MDS. The hematopoietic niche is a complex structure that contains a number of different cell types: multipotent mesenchymal stem cells (MSCs) and their progeny, a complex vascular network, nerve fibers, mature blood cells, etc. These cell types modulate HSC function and are frequently disrupted or even abnormal in the context of malignancies. It was shown that the components of the bone marrow niche and the HSC communicate in different ways, including: intercellular communication, cytokines, growth factors, mitochondria, and extracellular vesicles. In the context of the evolution of hematopoiesis during MDS development, the exchange/communication delivered by EVs and their miRNA content may influence genetic remodeling.

## 3. Role of EVs and miRNA as Diagnostic Tools

There is evidence that miRNAs promote survival and growth of malignant cells, notably in MDS and AML. Some miRNAs appear to be specifically deregulated in those hemopathies. Some studies have demonstrated the presence of tumor-specific miRNAs in plasma or serum, suggesting that those miRNA could be a potential biomarkers of cancers [61]. The main diagnostic issue with myelodysplastic syndromes is the necessity to perform bone marrow aspiration or bone marrow biopsy, which is quite invasive. The interest in using miRNAs in MDS as a diagnostic tool would be to use only peripheral blood to perform the diagnosis or to follow the disease without performing bone marrow aspiration.

Two studies have evaluated the utility of miRNA to differentiate MDS patients and healthy donors. Sokol et al. observed high levels of miR-222 and miR-10a and low levels of miR-146a, miR-150, and Let-7e in MDS [62]. Vasilatou et al. showed that let-7a, miR-17-5p, and miR-20a were overexpressed in low-risk MDS but underexpressed in high-risk MDS patients compared to healthy donors [63]. If the sensitivity and specificity of miRNA tools would be high enough to discriminate normal bone marrow from low-risk MDS, they would be complementary to next generation sequencing (NGS) of the recurrent gene mutations found in MDS.

Hypoplastic MDS and aplastic anemia are often difficult to diagnose as they are overlapped syndromes. Bone marrow biopsy and molecular biology testing cannot sometimes discriminate between both diagnoses. Giuduce et al. examined the possible diagnostic and prognostic values of exosomal miRNAs in aplastic anemia and myelodysplastic syndromes [64]. Very interestingly, by performing a screening of 372 miRNAs in exosomal plasma samples from patients at diagnosis, they observed 25 exosomal miRNAs that were uniquely present in aplastic anemia and/or MDS patients. Analysis of the targets of these 25 miRNAs reveals that they are involved in several biological functions, such as HSC differentiation, and intracellular functions, such as metabolism, cell survival, and proliferation (ERK5, PTEN, STAT3, VEGF signaling). In particular, the study reveals that fourteen miRNA are unique in MDS. Four other miRNA are common in MDS and aplastic anemia: miR-196a-5p, miR-196b-5p, miR-4267, and miR-378i. Clinical parameters (such as LDH, Hb level, response to immunosuppressive therapy) and progression-free survival (PFS) were correlated to miRNA expression levels in these patients. MiR-126-5p was negatively associated with response to immunosuppressive therapy in aplastic anemia. A higher level of miR-126-5p at diagnosis was associated with decreased PFS in patients with aplastic anemia (5.5 months versus 22.7 months).

In addition, other studies in the context of patients with 5q deletion syndrome report distinct miRNA expression profiles in CD34^+^ cells with overexpression of miR-34a and downregulation of miR-146a [62,65]. These miRNA expression profiles participate in the pathogenesis of 5q deletion syndrome.

Thol et al. analyzed mutations in miR-142 and associated miRs (miR-632 and miR-891) in a large cohort of 935 patients with AML or MDS. MiR-142 was mutated in MDS and in AML secondary to MDS [66]. Their study underlines the fact that miRNAs are involved in the regulation of normal hematopoiesis, as expected. Therefore, it is not surprising that a change in their expression may be responsible for the development of hemopathies. Over the past ten years, several large-scale studies of MDS-specific miRNA expression profiles (Table 1) have been published [67,68]. To better integrate the study of miRNAs into the prognostic process of MDS, studies first need to characterize in parallel the molecular abnormalities of MDS in order to make more detailed associations. This is a current limitation of these studies, which, however, have the potential for interesting clinical development incorporating miRNA profiles as a diagnostic tool.

## 4. Role of EVs and miRNA as Prognostic Tools and Prediction of Resistance to Therapy

Myelodysplastic syndrome prognosis is well characterized by IPSS-R score, which includes percentage of blasts, cytogenetics, and number of cytopenias. However, molecular biology has become increasingly used in MDS diagnosis and prognosis [71], and recently, a new prognostic score was published integrating molecular biology data: IPSS-M (https://doi.org/10.1056/EVIDoa2200008, accessed on 25 December 2022).

MiRNA have been evaluated as prognostic tools in MDS in numerous studies. For example, some miRNAs are correlated to shorter survival: miR-125a [70], let-7a [72], miR-194-5p [73], miR-22 [74], miR-661 [75], and others to disease progression: miR-422a, miR-617, miR-181a, miR-210, and miR-196-5b [72]. Furthermore, Chengyao et al. screened deregulated miRNA in the bone marrow sera of MDS patients. They demonstrated that expression of the miR-320 family was upregulated in MDS, and high expression of miR-320d was an independent prognostic factor for overall survival in MDS [69].

MiRNA could also be used as predictive factors of the response to therapy. Solly et al. [76] compared the expression of 754 miRNAs in cells of high-risk myelodysplastic syndromes resistant or sensitive to azacitidine. Azacitidine inhibits DNA methyltransferases, such as DNMT1. Seven miRNAs targeting the 3′UTR of DNMT1 are repressed in azacitidine-resistant cells and correlate with higher levels of DNMT1. The specific inhibition of endogenous miRNAs targeting DNMT1 increases the expression of DNMT1, thus, inducing resistance to azacitidine. Patients with low miR-126 had significantly lower response rates and higher relapse rates, as well as shorter progression-free survival, after treatment with azacytidine.

In a 5q deletion syndrome cohort treated with lenalidomide, transfusion independence and prolonged response correlated with an increase in miR-145 expression [77].

A major advantage of using miRNA in MDS prognosis or predication to drug resistance is that they can be isolated in blood without the necessity of blood marrow samples [78]. Nevertheless, the use of circulating and exosomal miRNAs in clinical practice requires technical improvement and standardization. It is very important, especially for the preparation of exosomes (differential ultracentrifugation or commercial kit), to extract the RNA and to normalize the data. For circulating miRNAs, normalization is now fairly consensual. It is performed using small nucleolar RNAs (snoRNAs), other miRNAS, or small RNAS. Methods such as GeNormPlus, NormFinder, and global mean of miRNA expression are used to define the better normalizer. Usually, two or three normalizers are necessary to perform normalization. However, in the case of exosomal miRNAs, there are no clear guidelines for data normalization. RNU6 is usually used for miRNA normalization, but it is not a reliable endogenous control for frozen samples [79].

## 5. Role of EVs and miRNA in the Chemoresistance

Currently, few data are available on the mechanisms induced by EVs and miRNAs leading to chemoresistance. We will, therefore, present in this section the most relevant results.

### 5.1. Drug Efflux

Efflux pump mechanisms perform important resistance to chemotherapeutic agents. For example, some tumor cells package P-glycoprotein (P-gp) in their secreted EVs, thus, allowing it to be delivered to nearby and distant cells where it modulates drug resistance. Vesicles released by drug-resistant cancer cells promote the incorporation of functional P-gp into drug-responsive cells that subsequently become drug-resistant. This observation has greatly contributed to understanding the molecular bases of drug resistance emergence in new therapeutic strategies targeting vesicle-mediated P-gp transfer [80]. More specifically in AML, EVs produced by leukemia cells can mediate the expression of drug efflux pump multidrug resistance protein -1 (MRP-1), which provides increased resistance to daunorubicin [81]. It is not clear how EVs increase MRP-1, but the mechanisms seem to involve miR-19b and miR-20a, which activate Pi3K/AKT signaling, leading to the overexpression of MRP-1 [82].

### 5.2. Signaling Pathway Modulation by microRNA

Extracellular miRNAs derived from cancer cells are transferred to cells in the bone marrow microenvironment where they will target immune response, angiogenesis, metastasis, and drug resistance, thus, promoting tumor development. The PTEN/PI3K/AKT pathway is a frequently found target of exosomal miRNAs linked to drug resistance. In hepatocellular carcinoma, exosomes from cells resistant to 5-fluorouracil (5-FU) are enriched in miR-32-5p. Overexpression of this miRNA inhibits PTEN expression and induces multidrug resistance. MicroRNA-21, miR-222, and miR-55 are associated with drug resistance in lymphoma, colorectal, ovarian, breast, and lung cancer. MiRNA-21 can modulate cancer cell chemosensitivity by targeting tumor suppressors such as PTEN [83] and FasL [84]. MiR-21 contributes to chemoresistance in acute myeloid leukemia by targeting PDCD4 and BTG2, which are pro-apoptotic genes [85]. Another study described exosomes from bone marrow stromal cells derived from AML patients enriched in miR-155 and miR-375 that play a role in the resistance to the tyrosine kinase inhibitor AC220 [86]. In addition, BMP-2 carried by extracellular vesicles and secreted by mesenchymal stromal cells induces survival of leukemia cells and promotes osteogenic differentiation of MSCs. These effects would be mediated by cell ER stress responses [87,88,89].

### 5.3. Protection of Leukemia Cells from Immunotherapy

Immune checkpoint molecules, including programmed cell death-1 (PD-1) and programmed cell death ligand-1 (PD-L1), play important roles in oncogenesis by maintaining an immunosuppressive tumor microenvironment. AML cells can release exosomes containing PD-L1, preventing T and NK-cell from immune recognition [90]. Exosomal PD-L1 can act as a decoy sequestering PD-L1 inhibitor, explaining partially the inefficacy of checkpoint inhibitors as anti PD-1/ PD-L1 in myeloid malignancies as AML and MDS [91].

## 6. Use of EVs and anti-miRNA as Treatment in MDS

### 6.1. Targeting miRNA

It is possible to silence aberrantly expressed miRNA using various nucleic acid analogues such as locked nucleic acids (LNAs) or peptide nucleic acids (PNAs) [92,93,94]. Nevertheless, current antimiR technologies are dependent on the efficiency of specific cell addressing. Cheng et al. [95] describe a novel antimiR delivery platform that targets the acidic tumor microenvironment. However, mimicking physiology may be the best way to deliver an antimiR [96,97], and several groups are working on specialized therapeutic exosomes. Recently, Kamerkar et al. [98] have engineered exosomes derived from fibroblast-like mesenchymal cells to carry siRNA to pancreatic cancer cells with a specific signal addressing the exosomes. In MDS, a chemically modified inhibitor of miRNA-21 promotes normal erythropoiesis and increases hematocrit [99].

Furthermore, other deregulated miRNAs could be potential targets for future therapy: miR-125a, which has already been demonstrated to play an important role in MDS CD34^+^ with a fine tuning of the NFKB pathway [70]; miR-148b, which is down-regulated in CML and can predict clinical behavior of patients stopping imatinib [100]; and miR-378, which is found deregulated in 5q deletion syndrome and in polycythemia vera [101,102]. MicroRNA-486-5p, by triggering innate immunity activation and oxidative stress in low-risk MDS, could also be an interesting target.

### 6.2. Extracellular Vesicles as Therapeutics

Finally, extracellular vesicles could be used for therapeutic applications. EV components derived from healthy donors are natural components with low immunogenicity, low toxicity, and high stability, which may be engineered to improve targeting and loading efficiency. MSC-derived EVs reduce ischemia and can be used in regenerative medicine [103].They can also attenuate inflammation or promote immune responses. Ongoing trials evaluate the efficacy of EVs isolated from dendritic cells to enhance immunotherapy in metastatic melanoma [104] or colon cancer [105].

EVs are also ideal delivery vehicles with stability in circulation and escape from phagocytosis contrary to mesenchymal stromal cells. They can deliver therapeutic biomolecules as nucleic acids. For example, EVs isolated from adipose-tissue-derived stromal cells, after transfection with a plasmid expressing miRNA-122, induced sensitization of hepatocellular carcinoma cells to sorafenib both in vitro and in vivo [106]. Anti-miRNA can also be delivered by EVs, as seen in a model of glioblastoma with anti-miRNA-9 [107].

Nevertheless, it is quite difficult to engineer in vitro extracellular vesicles. Bioinspired or biomimetic EVs, including artificially synthesized, EV-like NPs, EV-mimetic nanovesicles, or hybrid EVs were recently developed [108], but the main difficulty is to engineer a vesicle able to target a specific cellular subtype. To attain such aims, the surface of EVs has to be modified with molecules of interest to obtain reproducible results, while preserving vesicle integrity and activity. In this way, MSC are indicated as a potential new important tool for delivering anticancer agents, notably, by their ability to secrete extracellular vesicles [109]. Recently, O’Brien et al. have succeeded in engineering MSC to secrete EVs enriched with a specific miRNA using lentiviral transduction [110]. Moreover, MSC could be used to deliver active drugs through their secreted vesicles. For example, Pascucci et al. have demonstrated that MSC from breast cancer after priming with Paclitaxel can secrete enriched Paclitaxel EVs [111]. Those data suggest the possibility of using MSC as a factory to develop drugs with higher cell-target specificity.

In myelodysplastic syndromes, targeting the miRNA cargoed in EVs, which play a leading role in the pathophysiology of MDS, could be an interesting trail.

## 7. Conclusions

Unlike mRNAs, miRNAs embedded in extracellular vesicles are highly stable in the plasma samples, allowing the bone marrow material to be accessible by simple plasma analysis. An international IPSS-M score with molecular markers was recently presented and will be useful to orientate therapeutic strategies in myelodysplastic syndromes. With the emergence of non-coding RNA and the role of miRNA in the physiopathology of MDS, additional molecular-genetic markers could be integrated in the diagnosis and prognosis of MDS. Current data on the roles of miRNAs in MDS suggest that these molecules have the potential to become tools for diagnosis, prognosis, and treatment response predictive factors. The development of therapy targeting deregulated miRNA in the stromal hematopoietic stem cells by specific EVs could be a future way forward in the treatment of myelodysplastic syndromes.

## Figures and Tables

**Table 1 cells-12-00658-t001:** Most described miRNAs in myelodysplastic syndromes (MDS) physiopathology.

miRNA of Interest	Material Sampled	References
miR-10a	BM MNC, CD34^+^ cells, plasma EVS	[40]
miR-10b	BM MNC, CD34^+^ cells	[69]
miR-17-3p	BM MNC, PB	[62,63]
miR-17-5p	BM MNC, PB	[62,63]
miR-150	BM MNC, plasma EVs	[62]
miR-145	BM MNC	[49]
miR-126	BM MNC, CD34^+^ cells	[63]
miR-125a	BM MNC, CD34^+^ cells, plasma EVs	[70]
miR-222	BM MNC	[64]
miR-34a	CD34^+^ cells, plasma EVs	[64,65]
miR-99b	CD34^+^ cells, plasma EVs	[70]

BM MNC: Bone marrow mononuclear cells, EVs: Extracellular vesicles, PB: Peripheral blood.

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
