# Peer review of "Extracellular Vesicles and MicroRNA in Myelodysplastic Syndromes"

_cells, 2023, doi:10.3390/cells12040658_

Round 1
Reviewer 1 Report
In the present review Meunier and colleagues clearly address the role of extracellular vesicles and microRNA in myelodysplastic syndromes (MDS). They begin the review by explaining the biological role of extracellular vesicles and then moved to their potential clinical implications. This is an emerging field of research and I believe that this review allows the reader to clearly understand the future potential clinical implication of extracellular vesicles and microRNA in MDS.
Few additional comments:
• Line 76-77: could you clarify this first sentence of the paragraph?
• Line 149: the authors need to expand the legend of figure 1
• Line 353: the authors need also to cite the IPSS-M
(https://doi.org/10.1056/EVIDoa2200008)
• Line 354: please modify MiRNA into miRNA
• Line 468: the authors need to expand the legend of figure 3
Reviewer 2 Report
Meunier and colleagues present a thoughtful and useful review of literature pertaining to the role extracellular vesicles and their microRNA contents play in the pathogenesis of MDS. The first two thirds of the manuscript are really excellent: each point is explained and contextualized with the current literature, and the figures expand on the written text.
The last third of the manuscript, starting at point 2.4 (line 383), is a different story. It almost seems like the authors ran out of energy. There are spelling errors, the points are simply listed with a reference, and the conclusion is just a few sentences about miRNAs in the plasma. There should be some discussion in the conclusion addressing things like: what are the pitfalls and challenges that the field faces based on the published literature, and where do the authors see the field going in the near future?
Some minor points:
Line 50: The authors refer to two studies as "ancient." 1998 and 2012 (references 4 and 5) are older, but far from ancient
Line 141-142: What did Thery et al. in respect to the shape of exosomes in their study?
Figure 1 (line 148): are there known cell surface receptors that are involved in directing the phago/pinocytosis and fusion of microsomes and exosomes on the target/receiving cell?
Line 183: There are several ways in which miRNAs can suppress translation, it would be good to briefly touch on those
Line 203: "... resulting in AML." I am guessing that the authors are referring to the AML transition? It would be better to use that kind of language instead of "resulting in"
Line 214: Explain what 5q deletion is for the non-chromosome biology people reading this manuscript
Line 223: The authors need to pick a consistent notation style to communicate chromosome deletions or additions. For example, "5q deletion" vs "del5q"
Line 251: Cite the COSMIC database. They have a reference for you
Line 267-291: It is okay to summarize your own work in your review. This paragraph struck me as excessive self-citing. You can put it in context with what others have shown in the similar field, and compare the differences between what you see and what others see
Line 302: Along the same lines as above: you are showing your data in a review, which is inappropriate. You can draw a model and reference your paper, but raw data (like heat-maps and volcano plots) don't belong in a review. Figures 1 and 3 are great, do this one like those.
Line 321: Check the grammar
Line 383-the end: Check English spelling of the words and expand on these points. Chimioresistance = chemoresistance, for example
